# Risk Assessment of Transgender People: Development of Rodent Models Mimicking Gender-Affirming Hormone Therapies and Identification of Sex-Dimorphic Liver Genes as Novel Biomarkers of Sex Transition

**DOI:** 10.3390/cells12030474

**Published:** 2023-02-01

**Authors:** Roberta Tassinari, Alessia Tammaro, Gabriele Lori, Sabrina Tait, Andrea Martinelli, Luigia Cancemi, Paolo Frassanito, Francesca Maranghi

**Affiliations:** 1Center for Gender Specific Medicine, Istituto Superiore di Sanità, 00144 Rome, Italy; 2Experimental Animal Welfare Sector, Istituto Superiore di Sanità, 00144 Rome, Italy

**Keywords:** testosterone, estrogen, cyproterone acetate, masculinizing, feminizing, cytochrome P450, sex-specific genes

## Abstract

Transgender (TG) describes individuals whose gender identity differs from the social norms. TG people undergoing gender-affirming hormone therapy (HT) may be considered a sub-group of the population susceptible to environmental contaminants for their targets and modes of action. The aim of this study is to set appropriate HT doses and identify specific biomarkers to implement TG animal models. Four adult rats/group/sex were subcutaneously exposed to three doses of HT (plus control) selected starting from available data. The demasculinizing-feminizing models (dMF) were β-estradiol plus cyproterone acetate, at 0.09 + 0.33, 0.09 + 0.93 and 0.18 + 0.33 mg, respectively, five times/week. The defeminizing-masculinizing models (dFM) were testosterone (T) at 0.45, 0.95 and 2.05 mg, two times/week. Clitoral gain and sperm count, histopathological analysis of reproductive organs and liver, hormone serum levels and gene expression of sex-dimorphic CYP450 were evaluated. In the dMF model, the selected doses—leading to T serum levels at the range of the corresponding cisgender—induced strong general toxicity and cannot be used in long-term studies. In the dFM model, 0.45 mg of T represents the correct dose. In addition, the endpoints selected are considered suitable and reliable to implement the animal model. The sex-specific CYP expression is a suitable biomarker to set proper (de)masculinizing/(de)feminizing HT and to implement TG animal models.

## 1. Introduction

Transgender people (TG) represent a sub-group of population with a gender identity that differs from the assigned sex at birth [1]. They may show binary gender identities (i.e., TG men or TG women) or nonbinary; this last is an umbrella term entailing gender identities falling outside the binary gender frame. Some non-binary people identify themselves as predominantly male or female, though with aspects of the “other” gender (e.g., genderqueer) [2]. Recent estimations recorded that 25 million people were TG, but the number is still increasing; this poses serious problems in terms of how to include proper variables for an inclusive investigation of health concerns [3,4]. TG people often undergo gender-affirming/confirming hormone therapy (HT), but global medical regulatory agencies, e.g., European Medicines Agency or Food and Drug Administration, haven’t approved yet any hormonal agents or clinical protocols for TG medicine. Therefore, ‘off-label’ HTs are based on guidance from Endocrine Societies or similar [5]. The human HTs—usually administered lifelong—aim at bringing the sex hormone levels to the normal range of the corresponding cisgender and at the regression of secondary sex characteristics. For TG men, HT involves intramuscular or subcutaneous injection of testosterone (T); more recently, transdermal administration (patches or gel) of longer action has been proposed since it is suitable for long-term use. For TG women, HT usually includes β-estradiol (E2), alone or in combination with androgen-lowering drugs, administered transdermally, orally, or via injection. The dose levels and the time of administration change based on the means of E2 administration. The most common anti-androgen treatment in Europe includes cyproterone acetate (CPA) (50 mg daily) [6].

Indeed, scarce data are currently available on the potential HT impact on the health of TG people and on its potential long-lasting effects. In this context, toxicological issues should also be included, since one of the main goals of risk assessment is to characterize the chemical risks in potentially sensitive sub-population groups and to ensure the choice of proper safety factors [6]. In fact, TG people—as the general population—are daily exposed to food and environmental contaminants, among which endocrine disrupters share targets and modes of action with HT, making TG a sub-group of the population specifically susceptible and vulnerable to their effects. In this frame, specific animal models should be set and used to perform a reliable hazard identification for TG people undergoing HT [6].

In this context, the aim of the present study is the development of two innovative TG animal models—a demasculinizing-feminizing model (dMF) and a defeminizing-masculinizing model (dFM)—through the selection of suitable dose levels, time and way of administration of HTs on the basis of the current human therapies. Beyond the T serum level in the cisgender range, which is considered the best biomarker to evaluate the success of gender-affirming HT for both TG men and women, other specific functional and tissue biomarkers are studied and used to characterize the models, in particular: sperm count, clitoral dimensions, histopathological analysis of reproductive organs and liver, androgen receptor (AR) gene expression, E2 and luteinizing hormone (LH) serum levels. Specific attention is given to the liver: in fact, beyond its pivotal role in mammalian metabolism regulated by steroid hormones, a bidirectional link has been recently discovered connecting sex hormones and liver activity, due to the presence of both estrogen receptors (ERs) and ARs in the liver of both sexes [7]. Moreover, in this study, a panel of sex-specific liver cytochrome P450 isoforms (CYP450s) are selected to evaluate the plasticity of liver metabolism in relation with HTs and the potential application as biomarkers. Indeed, sexually dimorphic genes could be used to verify the trend/success/safety of HTs, also taking into account that the potential long-lasting effects of gender affirming HTs on the metabolism of TG men and women have not been established yet. For this purpose, the following CYPs have been studied: CYP2C11 is the major male-specific androgen 16α- and 2α-hydroxylase in the adult rat liver, induced at puberty following the neonatal androgenic programming in males but not females; CYP3A18, showing a similar profile, for which mRNA levels are 25-fold higher in the male rat liver compared to the female liver [8]; the steroid sulfate 15p-hydroxylase CYP2C12 is expressed in a female-specific way in adult rats—interestingly, CYP2C12 is present in both male and female rats at three weeks of age, but at puberty, it is increased only in females and fully suppressed in males [9]; CYP2C6 is a female-predominant isoform—the male rat liver expresses about 60% of the levels found in the female liver [10].

## 2. Materials and Methods

### 2.1. Ethical Approval

Animal studies were performed in accordance with Directive 2010/63/EU, the Italian Legislative Decree n. 26 of 4 March 2014 and the OECD Principles of Good Laboratory Practice (GLP). The study protocol was approved by the Italian Ministry of Health (authorization n° 806/2021-PR).

### 2.2. Experimental Design of the Animal Study

Sixteen young sexually mature Sprague-Dawley rats of both sexes (304 ± 13 g male rats and 190 ± 7 g female rats, 8/9 weeks old) were purchased from Envigo (Italy). They were housed two/cage, under standard laboratory conditions (22 ± 0.5 °C, 50–60% relative humidity, 12 h of dark-light alternation with 12–14 air changes per hour) with water and food (2018 Global Diet purchased from Mucedola, Milan, Italy) available ad libitum. In all cages, wood gnawing blocks were inserted for environmental enrichment and replaced weekly. After 1 week of acclimatization, same-sex rats were divided into two experimental groups (dMF and dFM, 4 rats/group), each composed of one control group (control male, CM, and control female, CF) and three dose levels of the selected HT.

The group size was calculated using G*Power software (latest ver. 3.1.9.7; Heinrich-Heine-Universität Düsseldorf, Düsseldorf, Germany) and starting from the serum levels of T, the key parameter to assess the success of HT, identified in previous in vivo studies after 2 weeks of therapy [11,12]. A power of 0.80 and a significance level (α) at 0.0167, equal to α = 0.05 was used to calculate the sample size considering the differences among the control and treated groups by applying the Mann–Whitney test with Bonferroni correction.

The calculated number of rats/groups:-for the dFM model, *n* = 4 based on T serum levels reported in Kinnear H.M. et al. [control group: 0.2 ± 0.3 and 0.45 mg of T group: 16 ± 5 ng/mL—mean ± standard deviation (SD)] [11];-for the dMF model, *n* = 3 according to Gomez A. et at. that report T serum levels of 1.901 ± 0.413 and 0.043 ± 0.023 ng/mL [mean ± standard error (SE)] in control and E2 plus CPA group (0.2 + 0.8 mg/kw bw day), respectively [12].

A group size of 4 rats was selected to even out the two rat models.

Two weeks of treatment and the subcutaneous injection as way of administration were chosen, considering the similarities with human HTs, the age correlation between rats and humans and the available data from the literature [6].

For both models, the doses were selected taking into account the main clinical guidelines used for TG people [5,13] and relevant data from the literature concerning rodent studies [11,12]. Moreover, a virtual calculator for dosage conversion between human and rat or mouse and rat was used (DoseCal, https://dosecal.cftri.res.in/index.php, accessed on 24 May 2022).

During the experimental procedures, all rats were monitored twice a day (at 8:30 a.m. and 4:00 p.m.) for general health conditions and potential aggressiveness due to HT administration. Body weight (bw) and food consumption were recorded two times a week. In the dFM model, the clitoral diameter of the rats was measured the first day of treatment and at the day of sacrifice by a precision digital caliper. Twenty-four hours after the last treatment, rats were anaesthetized with a gaseous solution of isoflurane and blood samples were collected by intracardiac puncture for the determination of serum hormones. Subsequently, animals were sacrificed by CO_2_ inhalation and necropsy and gross pathology were performed. Liver and reproductive organs (testes, ovaries and uteri) were excised and weighted. The right epididymis was used for sperm count analysis. For histopathological analysis, target tissues were immediately fixed in 10% buffered formalin, except for testes, fixed in Bouin’s solution. A portion of liver was flash-frozen in liquid nitrogen and stored at −80 °C for gene expression analysis.

#### 2.2.1. Demasculinizing-Feminizing Model (dMF)

Sixteen male rats were divided into four treatment groups (4 rats/group) receiving 200-μL subcutaneous injections of E2 (β-Estradiol 17-valerate, N. CAS: 979-32-8, Sigma-Aldrich, Milan, Italy) plus CPA (cyproterone acetate, N. CAS: 427-51-0 Sigma-Aldrich, Milan Italy) in sesame oil (N. CAS: 8008-74-0, ACROS, Monza (MI),Italy) or sesame oil alone five times a week for two weeks as follows:Control group (CM): sesame oil (vehicle);Dose 1 (D1M): 0.09 + 0.33 mg per dose;Dose 2 (D2M): 0.09 + 0.93 mg per dose;Dose 3 (D3M): 0.18 + 0.33 mg per dose.

#### 2.2.2. Defeminizing-Masculinizing Model (dFM)

Sixteen female rats were divided into four treatment groups (4 rats/group) receiving 100-μL subcutaneous injections of T (testosterone enanthate, N. CAS: 315-37-7 Sigma-Aldrich, Milan, Italy) in sesame oil or sesame oil alone twice a week (Monday and Thursday am) for two weeks as follows:Control group (CF): sesame oil (vehicle);Dose 1 (D1F): 0.45 mg per dose;Dose 2 (D2F): 0.95 mg per dose;Dose 3 (D3F): 2.05 mg per dose.

### 2.3. Sperm Count

Right *caudae epididymides* were excised, rinsed with D-MEM medium (Gibco Rodano (MI), Italy), transferred onto a Petri dish containing 1 mL D-MEM medium and minced with scissors. Epididymal pieces were fluxed through a Pasteur pipette to facilitate sperm extrusion. Sperm suspension was diluted up to 10 mL. Sperm were then counted by a Neubauer chamber under light microscopy (Nikon Eclipse Ts2, Amstelveen, The Netherlands) [14].

### 2.4. Blood Collection and Biochemical Evaluation of Hormones

Blood was always sampled from 9 to 10 am, with stratification across treatment groups, in order to reduce the potential impact of circadian rhythm and pulsatility. Blood samples were left to coagulate at room temperature for 1h, centrifuged twice for 15 min at 2000 rpm in a cooled bench-top centrifuge (Microlite Microfuge, Rodano (MI), Italy) and stored at −80 °C until use. Serum levels of all hormones were measured in the same analytical section by the following commercial ELISA kits of the same lot(s):-E2 Rat kit (RTC009R—BioVendor Brno, Czech Republic), LOD 2.5 pg/mL-T Mouse/Rat kit (RTC001R—BioVendor Brno, Czech Republic), LOD 2.5 pg/mL;-LH Rat Kit (ELK2367—ELK Biotechnology, Whuan, China), LOD 37.59 pg/mL

Each kit provided a standard solution of the hormone, and serial dilutions were prepared to derive a standard curve and define the range of linearity of each test. For all the analyses, the manufacturers’ instructions were followed. Each sample was assessed in duplicate, and absorbance was read at 450 nm on a VICTOR3 Multilabel reader (Perkin Elmer, Milan, Italy). The unknown hormone concentrations in samples were derived using the standard curve of each hormone using the software GraphPad Prism 6.0 (GraphPad Software Inc., San Diego, CA, USA).

### 2.5. Histological and Histomorphometrical Analysis

Immediately after the sacrifice, to avoid any possible post-mortem artefacts, the liver, ovary and uterus were fixed in 10% buffered formalin and the testes in Bouin’s solution and stored in 80% ethyl alcohol. They were dehydrated in a graded series of alcohol baths and embedded in paraffin by a tissue processor (Shandon Excelsior ES, Thermo Scientific, Rodano (MI), Italy). The 5-μm-thick histological sections were prepared using the Microm HM 325 (Thermo Scientific, Rodano (MI), Italy) and stained with hematoxylin/eosin for examination under light microscopy (Nikon Microphot FX, Amstelveen, The Netherlands) at various magnifications to evaluate the histopathological alterations. The scoring of the lesions was semi-quantitative, using a 5-point grading scale (0 to 4), taking into consideration the severity of the changes based on the criteria explained by Shackelfold C. et al. [15] and summarized as follows:

Grade 0: No change.

Grade 1 (+1): Minimal. Histological change that may be barely noticeable to changes considered so minor, small, or infrequent as to warrant no more than the least assignable grade (0–10%). For focal, multifocal or diffusely distributed lesions, this grade is used for processes where <10% of the tissue is involved. For hyperplastic/hypoplastic/atrophic lesions, this grade is used when the affected structure or tissue has undergone <10% increase or decrease in volume.

Grade 2 (+2): Mild. Histological change that is a noticeable but not prominent feature of the tissue. For focal, multifocal, or diffusely distributed lesions, this grade is used for processes where between 11% and 20% of the tissue is involved. For hyperplastic/hypoplastic/atrophic lesions, this grade is used when the affected structure or tissue has undergone between 11% and 20% increase or decrease in volume.

Grade 3 (+3): Moderate. Histological change that is a prominent feature of the tissue. For focal, multifocal, or diffusely distributed lesions, this grade is used for processes where 21–40% of the tissue section is involved. For hyperplastic/hypoplastic/atrophic lesions, this grade is used when the affected structure or tissue has undergone between 21% and 40% increase or decrease in volume.

Grade 4 (+4): Marked. Histological change that is an overwhelming feature of the tissue. For focal, multifocal, or diffusely distributed lesions, this grade is used for processes where 41–100% of the tissue section is involved. For hyperplastic/hypoplastic/atrophic lesions, this grade is used when the affected structure or tissue has undergone between 41% and 100% increase or decrease in volume.

The quantitative histomorphometrical analyses were performed on ovaries, uteri and testes by means of an image analysis system (Nis-Elements BR, Nikon Instruments, Amstelveen, The Netherlands) applied to an optical microscope (Nikon Microphot FX, Amstelveen, The Netherlands). Briefly, testis tubular diameters and the relative area of the seminiferous tubules and lumen were measured in 20 randomly selected tubules (10× objective); in the uterus, a cross-section was taken from the right uterine horn, 1 cm above the uterine bifurcation, and the ratio between the area of endometrium and myometrium as relative percentage of both uterine tissue components was calculated (2× objective) [16]. Moreover, the luminal epithelial cell height of the uterus was measured (64× objective). Ovarian classification of the different follicles was performed according to Fortune J.E. [17]. Using one of the largest sections in a central position of the ovary, primary and secondary follicles, corpora lutea, Graaf follicles and atretic follicles were counted in the whole ovarian section (2× objective) [16].

### 2.6. Gene Expression Analysis

10 mg of each liver sample were mechanically disaggregated by the Miccra D-1 homogenizer (ART-moderne Labortechnik, Müllheim, Germany) and total RNA content was extracted with the Norgen kit (Norgen Biotek Corp. Thorold, Canada) according to the manufacturer’s instructions. RNA quantity was assessed by Nabi Nano Spectrophotometer (MicroDigital Co. Ltd., Seoul, Republic of Korea), whereas integrity was evaluated by 1% agarose gel electrophoresis. All the samples met quality criteria (integrity, A260/A280 ≥ 1.8) to proceed with real-time PCR analysis. One microgram of total RNA from each sample was reverse-transcribed to cDNA using the Tetro cDNA Synthesis Kit (Quantace, Mumbai, India) according to the manufacturer’s instructions. Specific primers for CYP2C11, CYP3A18, CYP2C12, CYP2C6, AR and glyceraldehyde-3-phosphate dehydrogenase (GAPDH), as reference gene, were designed using the Primer-BLAST web application (www.ncbi.nlm.nih.gov/tools/primer-blast, accessed on 24 May 2022) and purchased from Metabion (Metabion International AG, Steinkirchen, Germany) as follows: CYP2C11 forward (fw) ACAGAGCTTTGGGAGAGGGA, reverse (rev) CCATGCAACACCACAAAGGG; CYP3A18 fw CAGCTGGGAAGGAAACTTGG, rev AGGCCATGAGAATAGGTCCC; CYP2C12 fw CTTGCCCCAAATGGTTTGTTG, rev GGTCAGGAACAAAAACAGCTC; CYP2C6 fw CCACGTTTATCCTGGGCTGT, rev GTGTCCAGGGACTGCTCAAA and AR fw TAGGGCTGGGAAGGGTCTAC, rev CCTCTGATGTGGGCTTGAGG.

The Excel TaqTM Fast Q-PCR Master Mix SYBR (SMOBIO Technology Inc., Hsinchu City, Taiwan) was used to perform real-time PCR assays, running reactions on a Bioer LineGene 9600 (Bioer, Hangzhou, China) with the following thermal program: one cycle at 95 °C for 20 s; 40 cycles at 95 °C for 3 s, 58 °C for 15 s and 72 °C for 15 s; one melting cycle from 55 to 95 °C to verify amplification products. Experiments were performed in duplicate on 96-well PCR plates. Threshold cycles were calculated by the LineGen 9620 PCR V.1.0 software (Bioer, Hangzhou, China). Data are expressed as ΔΔCt ± SD values for each target gene with control samples as calibrator and GAPDH as reference gene.

### 2.7. Data Analysis

Data management, data entry and statistical analyses were performed by a single operator using Microsoft Excel 2013. Data were analysed using the software JMP 10 (SAS Institute Inc., Cary, NC, USA). A non-parametric Kruskal–Wallis analysis was performed to analyse data, followed by post-hoc pairwise comparisons (Mann–Whitney test). Quantal data were analysed by a 2-way Fisher exact test to assess significant differences with respect to control groups. The Cochran–Armitage trend test was used to evaluate a dose-response trend. Differences between groups were considered significant if the *p*-value was < 0.05. The GraphPad Prism 6.0 software (Dotmatics, Boston, MA, USA) was used to design all graphics.

## 3. Results

### 3.1. General Toxicity, Sperm Count and Clitoral Gain

#### 3.1.1. Demasculinizing-Feminizing Model

No death, clinical effects or aggressive behaviour have been recorded. The bw gain, bw at treatment days 5, 8, 12 and 13, feed consumption, testis absolute weight and sperm count were significantly decreased in comparison to CM in all treatment groups (D1M, D2M and D3M) (Figure 1 and Figure 2; Table 1). Testis relative weight was significantly decreased in D2M and D3M in comparison to the CM group (Table 1). Liver absolute weight was significantly reduced in D3M and relative weight was significantly increased in D2M compared to the CM group (Table 1).

#### 3.1.2. Defeminizing-Masculinizing Model

No death, clinical effects or aggressive behavior have been recorded. The bw gain was significantly increased in D2F and D3F; bw at treatment days 12 and 13 was significantly increased compared to controls in D3F (Figure 3). Feed consumption was significantly increased in D3F compared to the CF group (Table 2). At necropsy, a dose-dependent increase of haemorrhagic ovaries and uteri, significant in D3F in comparison to the CF group (Table 2, Appendix A), was recorded. Clitoral gain was increased in all treatment groups, although not significantly (Table 2). Ovary absolute and relative weight showed a dose-dependent decrease, significantly in D2M and D3M in comparison to the CF group (Table 2). Uterus absolute weight was significantly decreased in D1F and D2F, whereas the relative weight was unaffected (Table 2). No treatment-related alterations were observed in absolute and relative liver weight (Table 2).

### 3.2. Biochemical Evaluation of Hormones

#### 3.2.1. Demasculinizing-Feminizing Model

T serum levels were significantly decreased in all treatment groups (D1M, D2M and D3M) in comparison to CM. E2 serum levels was statistically significantly increased in all treatment groups (D1M, D2M and D3M) in comparison to CM and in comparison to CF. No treatment-related alterations were observed in LH serum levels (Figure 4).

#### 3.2.2. Defeminizing-Masculinizing Model

T serum levels were dose-dependently and significantly increased in all treatment groups (D1F, D2F and D3F) in comparison to CF. No treatment-related alterations were observed in E2 serum levels. Serum levels of LH were significantly decreased in all treatment groups (D1F, D2F and D3F) in comparison to CF (Figure 5).

### 3.3. Histological and Histomorphometrical Analysis

#### 3.3.1. Demasculinizing-Feminizing Model

Testes showed a significant increase of tubule degeneration with tubular vacuolation, depletion of germ cells and disordered arrangement of the germ cell layer in all treatment groups (D1M, D2M and D3M) compared to the CM group (Figure 6; Table 3). Histomorphometrical analysis of testes showed a significant reduction of tubule area in the D3M group (Table 3). The liver showed a significant increase of sinusoidal dilatation (enlargement of the hepatic capillaries) and hepatocyte vacuolation in all treatment groups (D1M, D2M and D3M) compared to the CM group (Figure 7; Table 3).

#### 3.3.2. Defeminizing-Masculinizing Model

Uterus showed a dose-dependent increase of endometrial and/or myometrial hyperaemic vessels, significant in D3F in comparison to CF group. Histomorphometrical analysis showed a significant reduction of horn areas and myometrium areas in D1F and D2F groups and of lumen areas in D3F only. Endometrium areas were significantly reduced in all treatment groups (D1F, D2F and D3F). The ratio of endometrium and myometrium areas was significantly reduced in D3F, whereas it was significantly increased in D2F and D3F in comparison to the CF group. Luminal epithelium height was significantly decreased in the D2F and D3F groups (Table 4). The ovary showed a dose-dependent increase of hyperaemic vessels, significantly in the D3M group in comparison to the CF group (Appendix A). The number of primary and secondary follicles was significantly increased, and Graaf follicles were significantly reduced in the D3F group in comparison to the CF group (Table 4). The liver showed a significant increase of sinusoidal dilatation (enlargement of the hepatic capillaries) in all treatment groups (D1F, D2F and D3F) compared to the CF group (Figure 8; Table 4). On the contrary, hepatocyte vacuolation showed a significant decrease in D2F and D3F compared to the CF group (Table 4).

### 3.4. Gene Expression

#### 3.4.1. Demasculinizing-Feminizing Model

Male-specific CYPs were down-regulated in all treatment groups; in particular, CYP2C11 expression was significantly down-regulated in all treatment groups (D1M, D2M and D3M); CYP3A18 was significantly down-regulated in the D1M group in comparison to CM (Figure 9). No treatment-related alterations were observed in AR gene expression (Figure 9). Considering the female-specific CYPs, an up-regulation of CYP2C12 gene expression was observed in all treatment groups, statistically significant in D2M and D3M. The same dose levels induced up-regulation of CYP2C6 with borderline significance in D2M (*p* = 0.0518) (Figure 10).

#### 3.4.2. Defeminizing-Masculinizing Model

Female-specific CYP2C12 gene expression was down-regulated, significantly in D2F and D3F groups. No treatment-related alteration was present in CYP2C6 gene expression (Figure 10). CYPs specifically expressed in male rats were up-regulated in all treatment groups; in particular, CYP2C11 expression showed a dose-dependent up-regulation statistically significant in D2F and with borderline significance in D3F groups (*p* = 0.0518); CYP3A18 was significantly up-regulated in the D2F group in comparison to CF. No treatment-related alterations were observed in AR gene expression (Figure 9).

## 4. Discussion and Conclusions

In toxicological studies, the use of targeted animal models is of pivotal importance in order to obtain sound data for hazard identification of chemicals [18]. As an example, it is known that children are different from adults concerning chemical hazards due to different exposure scenarios, age-related metabolic capacity and biological susceptibility. The juvenile toxicity test—intended for hazard identification from a children-specific perspective and already used in paediatric drug development—is based on animal models at the peripubertal phase of life corresponding to childhood [19]. TG people undergoing HT showed unique features in terms of specific susceptibility and vulnerability to chemical contaminants; therefore, they need suitable animal models based on appropriate, novel biomarkers [14], and the development of these last represents the main aim of the present study.

Considering the dMF transition rat model, although no deaths were recorded during the treatment period, the selected doses of E2 plus CPA are shown to be too high and, consequently, the model cannot be implemented. In fact, as shown in Figure 11, several marked toxicological effects are recorded in all the selected doses even if the T serum levels are decreased, reaching the corresponding cisgender value in D2M and D3M. As expected, the feminizing HT increased the levels of E2 and caused marked testis alterations (weight, dimensions, sperm count and histopathology). In particular, the treatment impaired the tissue architecture and the different cellular types; indeed, Leydig cells in the adult are stimulated by the binding of LH to cAMP production, increasing the rate of cholesterol translocation into the mitochondria, cholesterol being the direct precursor of T. Interestingly, in the present study, the treatment did not alter the LH production.

At tissue level, the liver of male rats showed sinusoidal dilatation and hepatocyte vacuolation at all the doses. Several studies correlate E2 with hepatotoxicity. Ethynyl estradiol (a synthetic estrogen used as a contraceptive drug) caused liver effects (congestion, focal areas of haemorrhage, vacuolation of cytoplasm, distended sinusoids with dilated central veins) in female rats, and the extent and severity of the damage depends on the dose and the time of administration [20]. More recently, CPA was also demonstrated to be an aryl hydrocarbon receptor (AhR) agonist in mice and an AhR antagonist in human cells [21]. Indeed, in both male and female C57BL/10J mice, long-term exposure to 800 ppm of CPA in the diet induced hepatocyte hypertrophy and increased fat and glycogen [22]. Since the effects recorded in the present study are coherent with these data, they may serve as support to develop more targeted HTs, e.g., with reduced concentrations of CPA [23].

Two experimental studies using feminizing HT in rodents are available up to now: the Gomes A. et al. study focused on brain morphology after feminization therapy [12] and the Gusmão-Silva J.V. et al. study explored the effects of injectable steroid combination—frequently used by transwomen—on blood pressure and metabolic outcomes [24]. Although they are relevant for the health impact of HTs, none of them is specifically focused on the implementation of the rodent model for further applications.

Considering the masculinizing HT for female rats—the dFM model—the selected doses of T did not induce death or overt signs of toxicity; as expected, the bw gain was increased together with food intake due to T administration [25]. T serum levels were in the range of cisgender at all the doses; nevertheless, at the highest doses, uteri and ovaries with strong haemorrhage were noted, leading to identifying the D1F—0.45 mg—as the suitable T concentration to implement the model and for long-term studies (Figure 11). Clitoral areas were increased, although not significantly; in a previous study, three doses (0.225, 0.45 and 0.90 mg) of T enanthate were subcutaneously injected twice a week to post pubertal C57BL/6N female mice for 6 weeks, mimicking the HT for TG men [11]. Clitoral area was significantly increased at the highest doses, and the difference from the present data may be due to the shorter treatment period. Histopathological analysis of uteri showed atrophy in all treatment groups, compatible with T administration [26]. The alterations recorded in the ovary referred to increased haemorrhagic areas coherently with the observations at necropsy, and the delayed/reduced maturation of follicles in D3F. This last is again a typical effect of T administration in adult female mice [27]; indeed, the maturation of follicles in the ovary was not completely inhibited, as shown also by Yang M. et al., suggesting that T therapy did not deplete the ovarian reserve [11,27].

The LH suppression was already observed in mice treated with T, and E2 serum levels were decreased in the range of cisgender [11].

Liver alterations were similar but milder in comparison to those observed in the dMF model, above all concerning sinusoidal dilatation; interestingly, hepatocyte vacuolization was significantly reduced in the D2F and D3F groups in comparison to controls, indicating an apparent protective effect of T. Indeed, an interesting study of Uchida K. et al. showed that hepatic triglyceride accumulation diminished with sex maturation in male but not in female protein-restricted rats, since endogenous T reduces hepatic lipid accumulation. It might be hypothesized that masculinizing T treatment protected female rats from hepatic lipid accumulation [28], also providing an indirect hint to liver masculinization.

Other than the above-cited studies, there are some others that administered T to female rodents with different aims; as an example, Barteles C.B. et al. provided evidence that female mice produce normal, fertilizable eggs after six-week treatment with 400 μg T cypionate, whether T levels are low after a washout period or high during two active exposures [29]. In another study, female 12-week-old mice masculinized by subcutaneous injections of 0.9 mg T twice a week for three weeks, showed long-lasting epigenetic modification in the liver [30]. None of them provided specific data to implement a targeted animal model for TG people.

In the present study, CYP gene expression analysis provided interesting results: after just two weeks of HT in the dMF model, an apparent demasculinization in the expression of sex-specific CYPs occurred either for CYP2C11 or CYP3A18 (preferentially expressed in male rats and up-regulated by the treatment), and CYP2C12 (preferentially expressed in female rats and down-regulated) both in the D2M and D3M groups. The same was found for the dFM, where a reversion, in this case a defeminization, of sex-specific CYP expression with up-regulation of male-specific CYP3A18 and CYP2C11 and a down -regulation of female-specific CYP2C12 occurred, in particular in D2F and D3F. It should be also considered that the male and female hormonal profiles can be altered by several factors, including drug therapies, exposure to environmental chemicals and diseases such as diabetes and liver cirrhosis; these factors could consequently highly influence the expression of specific liver CYPs under hormone regulatory controls [9]. Nevertheless, in the present study, the marked shift of expression after a short (two weeks) exposure time might be considered as a biomarker of demasculinization/defeminization induced by the treatment [31]. In both the models AR expression was unaffected by the treatment, although it is known that its expression is sexually dimorphic and temporally patterned in the rodent liver [32].

In conclusion, as shown in Figure 11, concerning the dMF model, the selected doses of drugs—although leading to T serum levels in the range of the corresponding cisgender—have shown to induce strong general toxicity; for such reason, they cannot be used to implement the model and in long-term studies.

For the dFM model, the subcutaneous injection of 0.45 mg of T represents the correct dose to be administered for masculinizing female rodents also in long-term studies without appreciable adverse effects. In addition, the endpoints selected are considered suitable and reliable to implement the animal model.

CYP450 is a superfamily of membrane-bound enzymes expressed in almost all biological systems that play a crucial role in homeostasis and metabolism. Sex steroids regulate CYP enzymes in vitro and in vivo models; given that sex steroid concentrations are markedly increased or decreased among TG people undergoing HT, and that differential CYP expression may lead to different, sex-specific susceptibility, the potential role of sex steroids on drug-metabolizing enzyme expression and activity should be carefully studied [31]. In the present paper, for the first time, sex-specific CYP gene expression has been used as a biomarker supporting the set of proper (de)masculinizing or (de)feminizing HT in order to obtain a reliable animal model for TG people.

## Figures and Tables

**Figure 1 cells-12-00474-f001:**
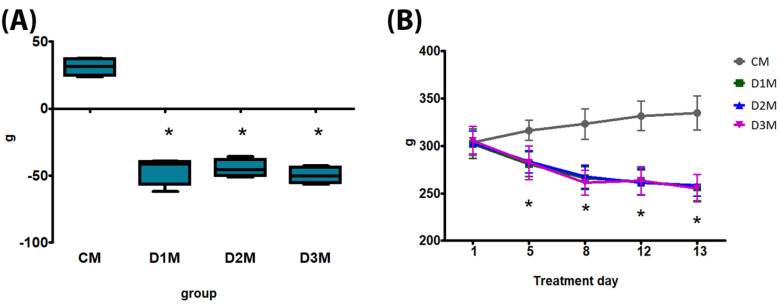
Body weight (bw) of male rats subcutaneously treated with different doses of estradiol valerate plus cyproterone acetate five times a week, for 2 weeks: CM: 0—sesame oil, D1M 0.09 + 0.33, D2M 0.09 + 0.93 and D3M 0.18 + 0.33 mg. Panel (**A**) bw gain and (**B**) bw at treatment days 5, 8, 12 and 13. Data are presented as mean ± standard deviation. Statistical significance: * *p* < 0.05 Mann–Whitney test.

**Figure 2 cells-12-00474-f002:**
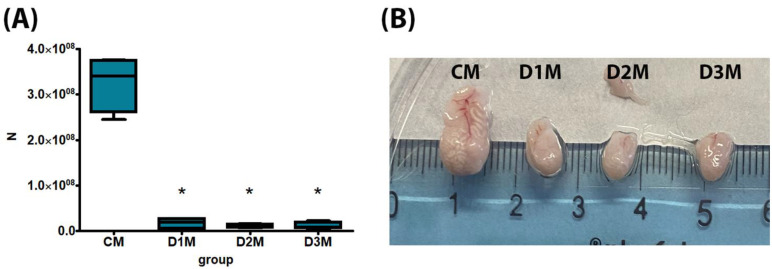
Sperm count of male rats subcutaneously treated with different doses of estradiol valerate plus cyproterone acetate five times a week, for 2 weeks: CM 0—sesame oil, D1M 0.09 + 0.33, D2M 0.09 + 0.93 and D3M 0.18 + 0.33 mg. Panel (**A**) sperm count and (**B**) picture of *caudae* epididymides. Data are presented as mean ± standard deviation. Statistical significance: * *p* < 0.05 Mann–Whitney test. *n*: sperm number.

**Figure 3 cells-12-00474-f003:**
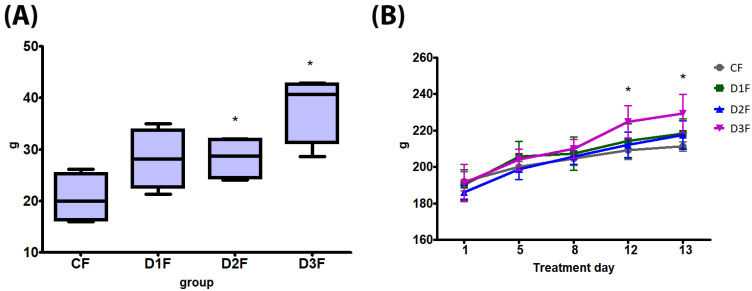
Body weight (bw) of female rats subcutaneously treated with different doses of testosterone enanthate two times a week, for 2 weeks: CF: 0—sesame oil; D1F: 0.45; D2F: 0.95; D3F: 2.05 mg. Panel (**A**) bw gain and (**B**) bw at treatment days 5, 8, 12 and 13. Data are presented as mean ± standard deviation. Statistical significance: * *p* < 0.05 Mann–Whitney test.

**Figure 4 cells-12-00474-f004:**
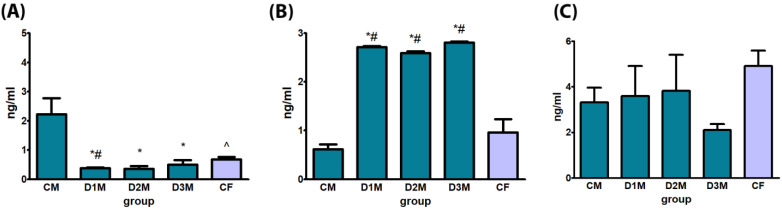
Biochemical evaluation of hormones by ELISA of male rats subcutaneously treated with different doses of estradiol valerate plus cyproterone acetate five times a week, for 2 weeks: CM 0—sesame oil, D1M 0.09 + 0.33, D2M 0.09 + 0.93, D3M 0.18 + 0.33 and CF (control female) 0 mg. Panel (**A**) testosterone, (**B**) estradiol and (**C**) luteinizing hormone. Data are presented as mean ± standard deviation. Statistical significance: * *p* < 0.05 “CM” vs “DM” groups; # *p* < 0.05 “CF” vs. “DM” groups; ^ *p* < 0.05 “CM” vs. “CF” groups Mann–Whitney test.

**Figure 5 cells-12-00474-f005:**
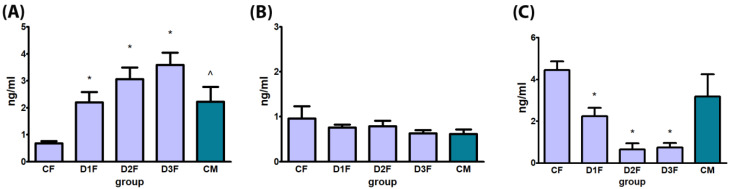
Biochemical evaluation of hormones by ELISA of female rats subcutaneously treated with different doses of testosterone enanthate two times a week, for 2 weeks: CF: 0—sesame oil, D1F: 0.45, D2F: 0.95, D3F: 2.05 and CM (control male) 0 mg. Panel (**A**) testosterone, (**B**) estradiol, (**C**) luteinizing hormone. Data are presented as mean ± standard deviation. Statistical significance: * *p* < 0.05 “CF” vs. “DF” groups; * *p* < 0. “CM” vs. “DF” groups; ^ *p* < 0.05 “CF” vs. “CM” groups Mann–Whitney test.

**Figure 6 cells-12-00474-f006:**
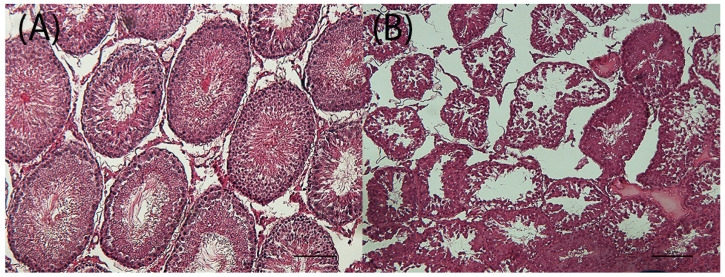
Testes of male rats subcutaneously treated five times a week, for 2 weeks with 0—sesame oil Panel (**A**) and 0.18 + 0.33 mg of estradiol valerate plus cyproterone acetate Panel (**B**). Tubule degeneration with germinal epithelium degeneration. Bar 10 μm (original magnification 10×; haematoxylin and eosin stain).

**Figure 7 cells-12-00474-f007:**
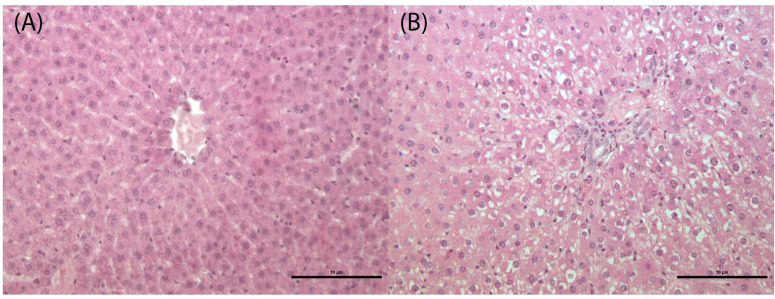
Liver of male rats subcutaneously treated five times a week, for 2 weeks with 0—sesame oil Panel (**A**) and 0.09 + 0.93 mg of estradiol valerate plus cyproterone acetate panel (**B**). Hepatocyte vacuolation. Bar 10 μm (original magnification 20×; haematoxylin and eosin stain).

**Figure 8 cells-12-00474-f008:**
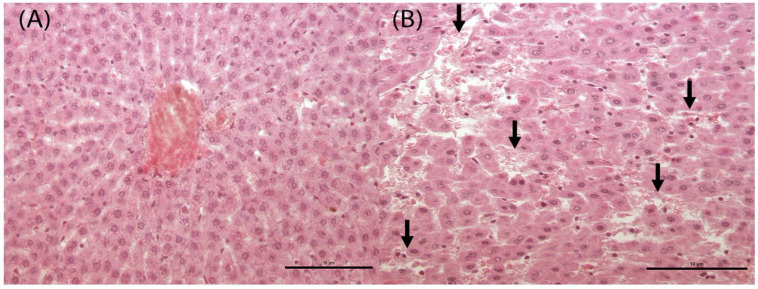
Liver of female rats subcutaneously treated two times a week, for 2 weeks with 0—sesame oil Panel (**A**) and 2.05 mg of testosterone enanthate Panel (**B**). Black arrows indicate sinusoidal dilatation (enlargement of the hepatic capillaries). Bar 10 μm (original magnification. 20×; haematoxylin and eosin stain).

**Figure 9 cells-12-00474-f009:**
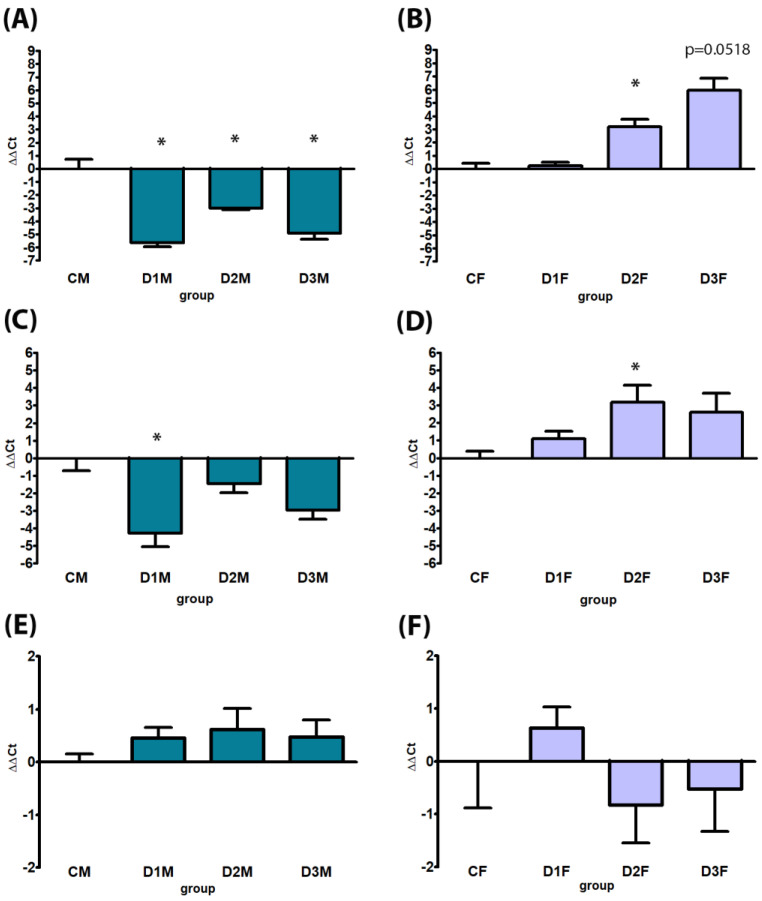
Comparison between gene expression analysis of sex (male)-specific genes CYP2C11 (Panels (**A**,**B**), CYP3A18 (panels (**C**,**D**) and AR (Panels (**E**,**F**) by real-time PCR in male and female rat livers. Male rats, panels (**A**,**C**,**E**), subcutaneously treated with different doses of estradiol valerate plus cyproterone acetate five times a week, for 2 weeks: CM 0—sesame oil, D1M 0.09 + 0.33, D2M 0.09 + 0.93, and D3M 0.18 + 0.33 mg. Female rats, panels (**B**,**D**,**F**), subcutaneously treated with different doses of testosterone enanthate two times a week, for 2 weeks: CF 0—sesame oil, D1F 0.45, D2F 0.95, and D3F 2.05 mg. Data are presented as mean ∆∆Ct values ± standard deviation, with control samples as calibrators and GAPDH as the reference gene. Statistical significance: * *p* < 0.05 Mann–Whitney test.

**Figure 10 cells-12-00474-f010:**
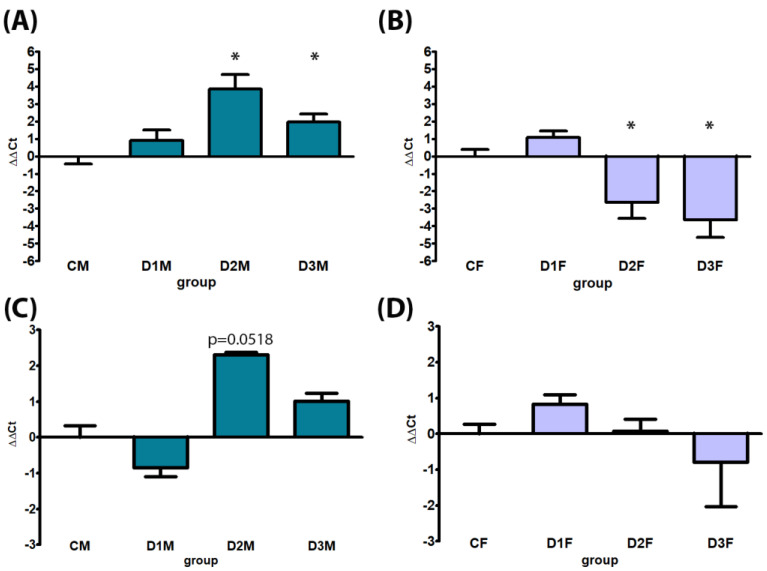
Comparison between gene expression analysis of sex (female)-specific genes CYP2C12 (Panels (**A**,**B**) and CYP2C6 (Panels (**C**,**D**) by real-time PCR in male and female rat livers. Male rats, panels (**A**,**C**), subcutaneously treated with different doses of estradiol valerate plus cyproterone acetate five times a week, for 2 weeks: C 0—sesame oil, D1M 0.09 + 0.33, D2M 0.09 + 0.93, and D3M 0.18 + 0.33 mg. Female rats, panels (**B**,**D**), subcutaneously treated with different doses of testosterone enanthate two times a week, for 2 weeks: CF: 0—sesame oil, D1F 0.45, D2F 0.95, and D3F 2.05 mg. Data are presented as mean ∆∆Ct values ± standard deviation, with control samples as calibrators and GAPDH as the reference gene. Statistical significance: * *p* < 0.05 Mann–Whitney test.

**Figure 11 cells-12-00474-f011:**
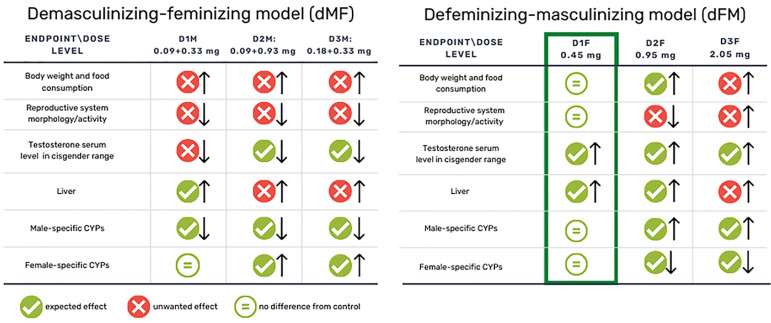
Main ‘solid endpoints’ used to establish the suitable dose for the implementation of the demasculinizing-feminizing (dMF) and defeminizing-masculinizing (dFM) models. Arrows indicate an ⇧ increasing or ⇩ decreasing in comparison to controls.

**Table 1 cells-12-00474-t001:** General toxicity data of male rats subcutaneously treated with different doses of estradiol valerate plus cyproterone acetate five times a week, for 2 weeks: CM 0—sesame oil, D1M 0.09 + 0.33, D2M 0.09 + 0.93 and D3M 0.18 + 0.33 mg. Statistical significance: * *p* < 0.05 Mann–Whitney test. *n*: number; SD: standard deviation.

	CM	D1M	D2M	D3M
*n*	4	4	4	4
Feed consumption (g, mean ± SD)	21.04 ± 0.08	13.48 ± 0.40 *	15.69 ± 0.77 *	14.97 ± 1.22 *
Testis absolute weight (g, mean ± SD)	3.49 ± 0.15	2.20 ± 0.50 *	2.16 ± 0.16 *	2.21 ± 0.21 *
Testis relative weight (mean ± SD)	1.05 ± 0.06	0.86 ± 0.22	0.83 ± 0.07 *	0.86 ± 0.07 *
Liver absolute weight (g, mean ± SD)	12.23 ± 1.17	10.03 ± 1.50	11.20 ± 0.96	10.14 ± 0.66 *
Liver relative weight (mean ± SD)	3.65 ± 0.26	3.91 ± 0.37	4.33 ± 0.24 *	3.97 ± 0.24

**Table 2 cells-12-00474-t002:** General toxicity data of female rats subcutaneously treated with different doses of testosterone enanthate two times a week, for 2 weeks: CF 0—sesame oil; D1F 0.45, D2F 0.95 and D3F 2.05 mg. Statistical significance: § *p* < 0.05 Fisher exact test; # *p* < 0.05 linear trend; * *p* < 0.05 Mann–Whitney test. *n*: number. SD: standard deviation.

	CF	D1F	D2F	D3F
*n*	4	4	4	4
Feed consumption (g, mean ± SD)	15.8 ± 0.2	16.8 ± 1.0	15.4 ± 0.5	17.9 ± 1.4 *
Haemorrhagic ovaries and uteri	0/4 (0%) ^#^	0/4 (0%)	1/4 (20%)	4/4 (100%) ^§^
Clitoral gain (mm, mean ± SD)	−0.12 ± 0.47	0.48 ± 0.43	0.24 ± 0.20	0.39 ± 0.36
Ovary absolute weight (g, mean ± SD)	0.17 ± 0.02	0.14 ± 0.12	0.12 ± 0.02 *	0.10 ± 0.01 *
Ovary relative weight (mean ± SD)	0.08 ± 0.01	0.07 ± 0.01	0.06 ± 0.01 *	0.05 ± 0.01 *
Uterus absolute weight (g, mean ± SD)	0.52 ± 0.07	0.36 ± 0.02 *	0.37 ± 0.03 *	0.41 ± 0.10
Uterus relative weight (mean ± SD)	0.21 ± 0.04	0.17 ± 0.01	0.17 ± 0.02	0.18 ± 0.05
Liver absolute weight (g, mean ± SD)	6.94 ± 0.14	7.36 ± 0.59	7.47 ±0.61	7.25 ± 1.45
Liver relative weight (mean ± SD)	3.25 ± 0.07	3.37 ± 0.25	3.48 ± 0.20	3.16 ± 0.63

**Table 3 cells-12-00474-t003:** Histopathological endpoints in target organs of male rats subcutaneously treated with different doses of estradiol valerate plus cyproterone acetate, five times a week, for 2 weeks: CM 0—sesame oil, D1M 0.09 + 0.33, D2M 0.09 + 0.93; and D3M 0.18 + 0.33 mg. Statistical significance: ^§^ *p* < 0.05 Fisher exact test; * *p* < 0.05 Mann–Whitney test. *n*: number; SD: standard deviation.

ORGAN/Observation		CM	D1M	D2M	D3M
	*N*	4	4	4	4
TESTIS: tubule degeneration with germinal epithelium degeneration					
	0	4			
	2		4		
	3			4	
	4				4
	Total Finding Incidence	0	4 ^§^	4 ^§^	4 ^§^
TESTIS: tuble area (μm^2^; mean ± SD)		422.3 ± 74.3	271.9 ± 147.1	264.8 ± 85.8	223.0 ± 30.7 *
LIVER: sinusoidal dilatation					
	0	4			
	1		2		
	2		2	4	4
	Total Finding Incidence	0	4 ^§^	4 ^§^	4 ^§^
LIVER: hepatocyte vacuolation					
	0	4			
	1		4		
	2			2	
	3			2	4
	Total Finding Incidence	0	4 ^§^	4 ^§^	4 ^§^

**Table 4 cells-12-00474-t004:** Histopathological endpoints in target organs of female rats subcutaneously treated with different doses of testosterone enanthate two times a week, for 2 weeks: CF 0—sesame oil, D1F 0.45, D2F 0.95 and D3F 2.05 mg. Statistical significance: ^§^ *p* < 0.05 Fisher exact test; ## *p* < 0.01 linear trend; * *p* < 0.05 Mann–Whitney test. *n*: number. SD: standard deviation.

ORGAN/Observation		CF	D1F	D2F	D3F
	*N*	4	4	4	4
UTERUS: endometrial and/or myometrial hyperaemic vessels					
	0	4	4	3	
	3			1	4
	Total Finding Incidence	0 ^##^	0	1	4 ^§^
UTERUS: total horn areas (μm^2^; mean ± SD)		60,651 ± 12,149	33,878 ± 3137 *	36,023 ± 8422 *	37,795 ± 11,903
UTERUS: lumen areas (μm^2^; mean ± SD)		3864 ± 3878	1355 ± 514	1683 ± 1364	1032 ± 248 *
UTERUS: myometrium areas (μm^2^; mean ± SD)		12,614 ± 1158	7660 ± 1396 *	7269 ± 2350 *	9464 ± 2730
UTERUS: endometrium areas (μm^2^; mean ± SD)		26,246 ± 3253	17,261 ± 1939 *	17,495 ± 3596 *	13,519 ± 5230 *
UTERUS: endometrium/myometrium areas (mean ± SD)		2.09 ± 0.28	2.31 ± 0.52 *	2.52 ± 0.60 *	1.44 ± 0.33 *
UTERUS: luminal epithelium height (μm; mean ± SD)		23.03 ± 1.56	18.10 ± 3.80	17.14 ± 2.23 *	17.60 ± 1.98 *
OVARY: hyperaemic vessels					
	0	4	4	2	
	2			2	4
	Total Finding Incidence	0 ^##^	0	2	4§
OVARY: follicular density (mean ± SD)		0.08 ± 0.0	0.11 ± 0.1	0.14 ± 0.0	0.15 ± 0.0
OVARY: primary + secondary follicles (*n*; mean ± SD)		28.5 ± 11.7	21.25 ±10.5	47.3 ±14.2	65 ± 6.8 *
OVARY: Graaf follicles (*n*; mean ± SD)		3.5 ±1.3	2.5 ± 1.0	2.5 ± 1.0	1.25 ± 0.5 *
LIVER: hepatocyte vacuolation					
0				4	4
1		4	4		
Total Finding Incidence		4	4	0 ^§^	0 ^§^
LIVER: sinusoidal dilatation					
	0	4			
	1		4		
	2			3	
	3			1	4
	Total Finding Incidence	0	4 ^§^	4 ^§^	4 ^§^

## Data Availability

Data is contained within the article or Appendix A.

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
