# Peer review of "Risk Assessment of Transgender People: Development of Rodent Models Mimicking Gender-Affirming Hormone Therapies and Identification of Sex-Dimorphic Liver Genes as Novel Biomarkers of Sex Transition"

_cells, 2023, doi:10.3390/cells12030474_

Round 1
Reviewer 1 Report
This manuscript describes a very interesting study dealing with the development of rodent models mimicking gender-affirming hormone therapies. In particular, to identify of sex-dimorphic liver genes to use as novel biomarkers of sex transition. To set appropriate hormone therapy doses and identify specific biomarkers to implement animal models, the authors treated both male that female rats with 3 doses plus control. For the feminizing model has been used estradiol, the major female sex hormone, plus anti-androgenic drug cyproterone acetate, while for masculizing model has been used testosterone. Toxicological endpoints as well as general toxicity, sperm count and clitoral gain have been evaluated together in the histophatological analysis of reproductive organs and liver, hormone serum levels (E2, T and LH), besides at the gene expression of sex-dimorphic CYP450.
The manuscript is well-written and the data support the conclusions the authors are suggesting. The work that has been done was very well thought and performed. Although a few editing needs to be done, the authors should define acronyms in the first place such as E2, T in abstract, introduction, etc., and these acronyms should be used them throughout the manuscript included the figure legend. Regarding to cyproterone acetate no acronym was used, therefore the authors should identify an acronym and use it in the manuscript, moreover on line 26 should be corrected cyprotenone acetate in cyproterone acetate. While, in figure 4 panel C and somewhat strange that there are no significant statistics, the authors should review the statistic. Finally, in figure 10 panel C should be replaced the data p=0.0518 reported shown above the D2M treatment column with the relative asterisk according the statistic value obtained.
Author Response
The Authors kindly thank the Referee for the interesting comments and the text has been carefully checked accordingly.
In the figure 4 panel C no statistically significant data are shown, that’s why no concurrent indications are provided.
In the Figure 10 panel C the asterisk is not present since the statistical significance has been considered for data with a p value < 0.05
Reviewer 2 Report
No sugggestions.
Author Response
The Authors kindly thank the Referee.
Reviewer 3 Report
The work by Tassinari et al. deals with the development of rodent models mimicking gender-affirming hormone therapies in humans. In particular, aim of the work was to set appropriate hormone therapy doses and to identify specific biomarkers. The authors investigated a series of biomarkers including sex-specific CYP expression in a complex framework of experimental studies. The final aim seems to be the development of a valuable animal model useful for transgender individuals undergoing gender-affirming hormone therapy.
The study is extremely original and relevant since the need of valuable animal models for the study of toxicity of therapies in use in transgender people appears as mandatory. The authors made humongous efforts to instruct a complex experimental study such as this and provide very useful information as concern the possible use of rat models. I find of particular interest the data on CYP450 and the findings dealing with dFM.
I have only two major concerns before the work could be considered for publication:
1. The role of the environment is certainly of great importance but, in my opinion, this is out of the scope of the present work. I can not understand the introduction of this matter in the work. Authors, please, explain or withdraw.
2. To help the reader in the understanding the mess of data provided, I would like to see a summary (a Table or a drawing) that could facilitate the identification of the main “solid points” obtained. Authors, please, consider, for example, a synopsis provided as a table for each treatment, dFM and dMF. I perfectly understand that this is a complex study but, just for this reason, an effort aimed at creating a graphical or synoptical document could help the reader and could represent a milestone in the field.
Minor points:
authors should better explain the pharmacological issues they considered: drug features, their use in the clinical practice and the known adverse events.
Seminiferous tubules: I would like to see a comment on Leydig cells injury. I think this is important. I see a complete derangement of these cells.
Author Response
The Authors thank the Referee for the productive comments that allow us to substantially improve the paper.
1.    The role of the environment is certainly of great importance but, in my opinion, this is out of the scope of the present work. I can not understand the introduction of this matter in the work. Authors, please, explain or withdraw.
In the frame of the hazard identification of chemicals – first step of the risk assessment process, the Authors intended to set animal models to test the impact of environmental chemicals on the health of transgender people (Tassinari & Maranghi 2021 doi: 10.3390/ijerph182312640). This is clearly indicated both in the Introduction and in the sentence ‘In toxicological studies, the use of targeted animal models is of pivotal importance to obtain sound data for hazard identification of chemicals’. In this respect, the animal models represented a key tool for the hazard identification of environmental chemicals specifically targeted to transgender people.
2.    To help the reader in understanding the mess of data provided, I would like to see a summary (a Table or a drawing) that could facilitate the identification of the main “solid points” obtained. Authors, please, consider, for example, a synopsis provided as a table for each treatment, dFM and dMF. I perfectly understand that this is a complex study but, just for this reason, an effort aimed at creating a graphical or synoptical document could help the reader and could represent a milestone in the field.
An additional Figure that summarizes the relevant endpoints used to set the models has been provided. Also a graphic abstract has been added.
Minor points:
Authors should better explain the pharmacological issues they considered: drug features, their use in the clinical practice and the known adverse events.
As explained in the paper, for both models, the drugs and their doses were selected taking into account the main clinical guidelines used for TG people and relevant literature data concerning rodent studies. The issues have been described in detail in (Tassinari & Maranghi 2021 doi: 10.3390/ijerph182312640). In the present paper, focused on animal models, it appeared to be of little relevance to discuss clinical practices.
Anyway, such issue has been better described in the ‘Introduction’
Seminiferous tubules: I would like to see a comment on Leydig cells injury. I think this is important. I see a complete derangement of these cells.
As expected, the testes appeared quite completely deranged. It was quite impossible to evaluate the architecture of the tissue (please see the pictures). A comment on Leydig cells has been included in the Discussion